# The Impact of Environmental Regulation on Hebei’s Manufacturing Industry in the Global Value Chain

**DOI:** 10.3390/ijerph20042933

**Published:** 2023-02-08

**Authors:** Fangmiao Hou, Wei Su, Shiyi Cheng, Chengliang Wu, Yuguo Lin

**Affiliations:** School of Economics and Management, Beijing Forestry University, Beijing 100083, China

**Keywords:** environmental regulation, position in the global value chain, manufacturing industry in Hebei province

## Abstract

In order to tackle increasingly serious environmental problems, China has been promoting the development of a green economy and guiding the green transformation of various regions and industries through environmental regulation in recent years. By participating in international trade, Hebei Province has been embedded in the global value chain. However, Hebei’s involvement in the high-energy-consuming and polluting manufacturing sector and its lower position in the global value chain have led to serious environmental issues. In practice, the government has promulgated environmental regulations to restrict economic activities of enterprises. What role does environmental regulation play in Hebei’s manufacturing industry’s participation in the global value chain? In order to explore the impact of environmental regulation on Hebei’s manufacturing industry in the global value chain, this paper constructs a fixed-effect econometric model based on the panel data of the embedding level of the value chain of 12 manufacturing sectors in Hebei Province. The research results show that: first, the R & D capacity of the manufacturing industry in Hebei Province still needs to be improved. Second, environmental regulation has promoted the global value chain position of Hebei’s 12 manufacturing sectors. Third, environmental regulation will show obvious heterogeneity to manufacturing industries with different capital intensities and different pollution levels. The impact of environmental regulation on the manufacturing industry with different intensities is different. Therefore, the government should formulate targeted environmental regulation to promote the position of Hebei’s manufacturing industry in the global value chain, such as further improving environmental regulation and increasing the intensity of environmental regulation and increasing the investment of human capital, and cultivating innovative talents.

## 1. Introduction

Since the reform and opening up in 1978, China has vigorously developed the manufacturing industry and gradually embedded itself into the global value chain (hereinafter referred to as GVC) with the advantage of foreign investment, labor, land, and domestic market [1]. In 2021, the added value of China’s manufacturing industry has reached 31.4 trillion Yuan, accounting for 27.4% of the GDP. Although the added value of China’s manufacturing industry ranks first in the world, most of the products rely on cheap labor and natural resources. This mode does not raise the position of the manufacturing industry in the GVC, but brings about high energy consumption, waste of resources, and high pollution [2].

The Beijing-Tianjin-Hebei region is highly interrelated in location, industries, and potential for regional development. In order to promote coordinated regional development and build a high-quality development pattern, there is an imperative need to vigorously develop the Beijing-Tianjin-Hebei region [3,4,5,6,7]. Beijing-Tianjin-Hebei is an important region for China’s economic development. Its economic structure is mainly dominated by high-energy consumption manufacturing industries, especially heavy industries in Hebei Province. From 2007 to 2019, the average annual proportion of the secondary industry was 49.69%, and the average annual proportion of the tertiary industry was 39.12%. Beijing-Tianjin-Hebei has made some cooperation in economy, industry, infrastructure and environmental resources. Among them, Beijing and Tianjin have mainly developed knowledge-intensive service industries, while Hebei mainly develops manufacturing industries with low-end products, which has brought serious environmental pollution problems [8]. Therefore, it is very important for Hebei Province to promote the structural adjustment, transformation, and upgrading of the manufacturing industry through various policies including environmental regulation [9]. In practice, the government restricts some economic activities of enterprises by formulating and promulgating policies and regulations to achieve the purpose of environmental protection. What role does environmental regulation play in Hebei’s manufacturing industry’s participation in the global value chain? Thus it is important to clarify the impact mechanism of environmental regulation on the GVC position of the manufacturing industry in Hebei Province, in order to maximize the role of environmental regulation according to local conditions, and promote the upgrading process of the manufacturing industry in Hebei Province faster.

The current literature is mainly about the effects of environmental regulation on the manufacturing industry and its subsectors at the national or regional level, or the overall manufacturing industry at the regional level, and there are fewer studies about the impact of environmental regulation on the GVC embedding of the manufacturing industry and its subsectors at the provincial level [10,11,12,13,14,15,16]. Therefore, there is an urgent need to bridge the gap and conduct studies on the position in the GVC of the 12 manufacturing sectors in Hebei Province, to verify whether they are affected by environmental regulations and how much they are affected. The structure of the paper is as follows: The second part discusses the impact mechanism of environmental regulation on the position of manufacturing GVC, and the third part is the research methods and data used in this paper. The fourth part is the empirical research results while the fifth part is the conclusion and policy implications.

Compared with the existing research, the innovation of this paper may lie in:

First, most scholars study the participation of manufacturing industries or some sectors in the global value chain at the national level, the literature are limited regarding the impact of environmental regulation on the sectors of manufacturing at the regional level [1,2,17,18]. This paper selects Hebei Province’s specific manufacturing sectors as the research subject, and discusses the impact of environmental regulation on their position in the global value chain, including manufacturing sectors with different capital intensities and with different levels of environmental pollution as new influencing factors in order to demonstrate what impact of environmental regulation will have on specific manufacturing sectors.

Second, previous scholars studied the participation of the manufacturing industry in the global value chain, mostly from the national or regional level, using the world input-output table [12,13,15]. This paper nested the world input-output table and China’s national input-output table, built an embedded world input-output table model, measured the global value chain status index of manufacturing industry segments in Hebei Province, and studied its development status.

## 2. Theoretical Mechanisms

Based on the existing theory, environmental regulations affect the position of the manufacturing industry in the GVC mainly from two aspects: passive compliance and benign development.

The analysis of passive compliance is based on the externality theory and resource allocation distortion theory [19,20]. The core idea of the above-mentioned theories is that environmental regulations make the manufacturing industry spend more on pollution control, such as the purchase of environmentally friendly equipment and the improvement of production processes leading to higher costs. Moreover, pollution taxes, charges, environmental certification, and other fees will lead to an increase in operating costs. In order to adapt to the change in cost, enterprises need to change their allocation of production factors in time, and optimize the efficiency of resource allocation. Therefore, the price of production factors has also changed [21], which has two impacts on the GVC position of manufacturing enterprises. On the one hand, it may reduce competitiveness in trade due to higher costs. On the other hand, it has squeezed the profits of enterprises and reduced their production capacity to some extent. In the short run, enterprises do not have enough funds to complete the optimal allocation of production factors and improve their position in GVC [22,23,24,25,26,27,28,29].

The analysis of benign development is based on Porter’s hypothesis and resource allocation distortion theory [20,30]. The core idea is that under environmental regulation, the production cost of manufacturing enterprises will rise, which will hinder the development of enterprises in the short run. However, enterprises affected by environmental regulations would increase their own R & D investment in order to achieve long-term development in the future and improve productivity and increase added value through its technological innovation [31]. Most scholars mainly hold the above two views, that is, when the government conducts environmental regulations on enterprises, it will have positive and negative effects. The final impact of environmental regulation on the regulated enterprises depends on the extent of both the positive and negative effects. The regulated enterprises are mainly affected negatively in the early stage of the enactment of environmental regulation, while the positive effects can be more effective in the long run because R & D of enterprises have made certain achievements, resulting in high added value and high productivity brought by R & D and innovation far exceeding the cost under the influence of environmental regulation. Therefore, although environmental regulation will have positive and negative effects on the regulated enterprises, it will eventually improve the technological level of the enterprises’ productivity and be in an advantageous position in international competition. At the same time, it will also bring the high added value of products, increase profits and improve enterprise capacity. By improving productivity and avoiding “low-end locking”, the position of the industry in the GVC can be improved [8,11,32].

According to the above analyses, environmental regulation will have both negative and positive impacts on the GVC division’s position of the manufacturing industry, and its final impact is uncertain. With regard to environmental regulations, enterprises must take positive measures to adjust their future strategies and optimize the allocation of internal and external resources to enhance their competitive advantages. Therefore, the positive impacts of environmental regulations on manufacturing can make up for their negative impact. Therefore, Hypothesis 1 is put forward: when other conditions remain unchanged, environmental regulations can promote the position of Hebei’s manufacturing industry in the GVC.

Moreover, different manufacturing sectors will be constrained by their heterogeneity facing environmental regulations according to the different capital intensity and pollution levels. Environmental regulations have great impacts on pollution-intensive manufacturing sectors and capital manufacturing-intensive sectors. As the polluting-intensive manufacturing sector is an important source of pollution, it is subject to greater control. Therefore, after the implementation of relevant environmental regulations, production costs will increase rapidly, and the prices of some factors will also rise. However, in order to develop the green economy in the future, enterprises will increase investment in environmentally friendly technology and adjust their own resource allocation. Therefore, based on the characteristics of different manufacturing sectors, Hypothesis 2 is proposed: environmental regulations will have different impacts on industries with heterogeneity.

The theoretical mechanism diagram is shown in Figure 1.

## 3. Methods and Materials

### 3.1. Methods

In order to analyze the position of Hebei’s manufacturing industry in the GVC, an embedded world input-output table (IRIOT-WIOD) needs to be constructed. The division of sectors of the World Input and Output Table is almost the same as that of the ADB database, which is both in accordance with the International Standard Industry Classification (ISIC) version 3, so the two versions of the input and output table can be combined.

The steps of constructing an embedded world input-output table are as follows: The first step is to merge the different sectors. According to the national economic industry codes of ISIC version 3 and ISIC version 4 in 2002 and 2011, the sectors are merged into 21. The second step is to merge the relevant countries. Due to the economic integration of the European Union, this paper will merge data of countries in the European Union into one. At the same time, the data from Japan and Korea are also merged for the convenience of calculation. So, we obtain countries including five parts such as China, the United States, the European Union, Japan, and South Korea and other countries are obtained. The third step is to embed the table. Split the overall data of China based on the regional input-output table, and adjust the data in the world input-output table to obtain the initial values of the parameters of China’s different provinces. In accordance with the corresponding proportion of China’s interregional input-output table, we determine the flow of intermediate goods, the final use, and the total value of intermediate inputs embedded in the world input-output table. The proportion of each province in 21 sectors is calculated through the EPS (Easy Professional Superior) data platform, and the corresponding positions embedded in the world input-output table are allocated in proportion with the import and export data of the world input-output table. The fourth step is data balance. Through the double proportion fitting algorithm, the total output is used as the control value for horizontal balance, and the intermediate input is used as the control value for column balance. Finally, the embedded input-output table is obtained.

### 3.2. Method of Calculating the Embedded Degree of GVC

The decomposition methods of GVC mainly include the HIY method, WWP method, KPWW method, KWW method and WWZ method. The WWP method further expands the measurement method of the disposal specialization index, and is no longer limited to a single country, but includes multiple countries. The WWZ method (Wang et al., 2015) further expanded the scope of the country’s various sectors and included bilateral trade, which decomposes a country’s export amount at multiple levels. On the basis of theoretical analysis, the WWZ method is used to decompose the added value of the export volume of the manufacturing sectors in Hebei Province. By dividing the trade into domestic and foreign value added [33], Equations (1)–(3) is used to calculate the forward participation, backward participation and GVC position index of the manufacturing sector in province level.
(1)$${\mathrm{gvcparf}}_{\mathrm{ir}}=\frac{{\mathrm{IV}}_{\mathrm{ir}}}{E_{\mathrm{ir}}}$$
(2)$${\mathrm{gvcparb}}_{\mathrm{ir}}=\frac{{\mathrm{FV}}_{\mathrm{ir}}}{E_{\mathrm{ir}}}$$
(3)$${\mathrm{gvcpos}}_{\mathrm{ir}}=\ln\left(1+\frac{{\mathrm{IV}}_{\mathrm{ir}}}{E_{\mathrm{ir}}}\right)-\ln\left(1+\frac{{\mathrm{FV}}_{\mathrm{ir}}}{E_{\mathrm{ir}}}\right)$$

In the above equations, $E_{\mathrm{ir}}$ represents the export volume of r industry in i province, ${\mathrm{IV}}_{\mathrm{ir}}$ represents the added value of indirect exports of r industry in i province, and ${\mathrm{FV}}_{\mathrm{ir}}$ indicates the foreign added value in the export of r industry in i province, ${\mathrm{gvcparf}}_{\mathrm{ir}}$ indicates the forward participation of r industry in the GVC in i province, and ${\mathrm{gvcparb}}_{\mathrm{ir}}$ indicates the backward participation of r industry in the GVC in i province, and ${\mathrm{gvcpos}}_{\mathrm{ir}}$ represents the global value chain division position of r industry in i province. Equation (1) measures the GVC forward participation of a certain industry based on the forward industry association in a certain province. Equation (2) measures the backward GVC participation of a certain industry in a certain province. Equation (3) measures the GVC position of the different manufacturing sectors in a certain province.

### 3.3. Selection and Definition of the Model’s Variables

The data used in this study are panel data from 12 sectors of Hebei manufacturing industry excluding wood processing and furniture manufacturing in 2002, 2007, 2010, 2012 and 2015. Based on the analysis of the mechanism that affects the GVC position in manufacturing sectors, the corresponding variables are selected.

“Regulate” is the core explanatory variable, indicating the intensity of environmental regulation. Environmental regulations are mainly divided into formal environmental regulations and informal environmental regulations [34]. The measurement methods of environmental regulation mostly adopt single indicator method or comprehensive indicator methods [3,25,35,36]. As the research is mainly about Hebei Province’s manufacturing sectors, variables of the environmental regulation selected in Hebei Province belongs to formal environmental regulations, and the measurement method is a single indicator method based on the environmental regulations from a cost perspective [8]. The specific methods are as follows:(1)Divide the investment in industrial pollution HZ of Hebei Province in 2002, 2007, 2010, 2012 and 2015 by those of the total investment in industrial fixed assets I, and obtain the ratio HR = HZ/I;(2)Multiply the fixed asset investment I and HR of the 12 manufacturing sectors in Hebei Province to obtain the investment in pollution control HC of the manufacturing sectors in Hebei Province;(3)From the perspective of output, the environmental regulation is measured by dividing the investment HCi in industrial pollution control of 12 manufacturing sectors in Hebei Province by their sales value. From the perspective of cost, the ratio of HCi divided by the cost of 12 manufacturing sectors in Hebei Province in industrial pollution control investment is used to measure environmental regulation (Li, 2010; Zhang et al., 2011).

In order to explore whether environmental regulation has an impact on the GVC position of manufacturing sectors in Hebei Province, we also need to control other factors. Therefore, the control variables selected in this part include human capital (Hr), industry size (Size), industry development level (Devp), R & D ability (Rd), and openness to the international market (Market).

The descriptions of variables and data sources are shown in Table 1.

### 3.4. Econometric Model

In order to solve the problem of missing variables, a two-way fixed effect (Two -way FE) model incorporating time and industry is used to empirically analyze the impact of environmental regulation on the GVC position of Hebei’s different manufacturing sectors [3]. The model is shown in Equation (4).
(4)$${\mathrm{Position}}_{\mathrm{it}}=\alpha_{0}+\alpha_{1}{\mathrm{regulate}}_{\mathrm{it}}+\alpha_{2}X_{\mathrm{it}}+{\mathrm{year}}_{t}+\mu_{i}+\varepsilon_{\mathrm{it}}$$
where i stands for industry and t for year; ${\mathrm{position}}_{\mathrm{it}}$ is the dependent variable, indicating the position in the GVC of Hebei’s manufacturing sector; ${\mathrm{regulate}}_{\mathrm{it}}$ is the core independent variable, indicating the intensity of environmental regulation; $X_{\mathrm{it}}$ is a series of control variables, including human capital, industry scale, industry development level, scientific research ability and openness; ${\mathrm{year}}_{t}$ is time fixed effect, $\mu_{i}$ is the fixed effect of the industry, and $\varepsilon_{\mathrm{it}}$ is a random error term.

## 4. Results

### 4.1. Descriptive Analysis

It can be seen from Table 2 that the standard deviation of human capital, industry scale, and industry development level is large, which indicates that there are large differences among the influencing factors of manufacturing sectors in Hebei Province.

### 4.2. Measurement Results

The absolute values of variables are less than 0.8 through the calculation of the Pearson correlation coefficient, which indicates that there is no obvious correlation between variables and the selection of variables is reasonable. According to Hausman test, the random model is rejected at a significant level of 1%. Therefore, a fixed effect model was adopted and dummy variables for each year were constructed, obtaining that the original hypothesis that time effect should be rejected. Therefore, time fixed effect was included.

The regression results of the benchmark model are shown in Table 3. Table 3 contains two perspectives, i.e., output and cost. Each perspective contains two models. Model 1 is the impact of environmental regulation on the GVC position of manufacturing sectors in Hebei Province when control variables are not introduced, followed by the estimation with control variables. The results of model 1 show that environmental regulation contributes to the improvement of the position of Hebei’s manufacturing sectors in the GVC, and it is significant at the level of 1%. After the introduction of control variables, the coefficient of environmental regulation impact has changed, but it is still significantly positive at the level of 1%, indicating that the increase of environmental regulation will help improve the position of Hebei’s manufacturing sectors in the GVC. The empirical results show that strict environmental regulation policies will promote the R & D and development capabilities of regulated enterprises. Through the reasonable transformation of technological achievements, productivity is improved and the added value of products is also increased. Thus, manufacturing enterprises have stronger competitiveness in international trade, and gradually improve their position in the GVC. Therefore, H 1 is verified, that is, when other conditions remain unchanged, the implementation of environmental regulations has promoted the GVC position of the 12 manufacturing sectors in Hebei Province.

The estimation with control variables shows that: (1) Human capital contributes to the promotion of the GVC position of 12 manufacturing sectors in Hebei Province. It may be that the advanced human capital provides some technical support for the development of the industry, but there are also negative effects of simple works on the transformation and upgrading of the industry. (2) The scale of the industry does not significantly affect the promotion of the 12 manufacturing sectors in Hebei Province in the GVC. The possible reason is that the scale of the manufacturing industry in Hebei Province is too large and has high difficulty in adjusting the factor endowment structure in a short time, which is not good for the transformation and upgrading of the industry and the promotion of its position in the GVC. However, environmental regulation promotes the manufacturing sectors in Hebei Province to accelerate the adjustment of resources and industrial structure, so it has a negative impact, but not significant. (3) The level of industry development has a restraining effect on the promotion of the position of the 12 manufacturing sectors in Hebei Province in the GVC. It may be that Hebei Province mainly relies on labor-intensive and capital-intensive manufacturing sectors to promote its development, which is characterized by low technology content and low development level. Therefore, it is difficult to complete scientific and technological R & D innovation, production factor structure adjustment, and industrial structure transformation in a short time. (4) The R & D ability has a restraining effect on the promotion of the position of the 12 manufacturing sectors in Hebei Province in the GVC. This may be due to the high cost of R & D, long time span, and low conversion rate. It is difficult to improve the position of the GVC in a short time. (5) The degree of openness to the international market has a significant inhibiting effect on the promotion of the position of the 12 manufacturing sectors in Hebei Province in the GVC. It may be that the high openness of the market has brought about relatively strong competition, thus hindering its position in the GVC.

In order to avoid endogeneity caused by two-way causality, this paper uses an endogeneity test to conduct a robustness test, taking environmental regulation as an endogenous variable, and selecting instrumental variable used data lagging behind one period. The weak instrumental variable test is used to determine whether the instrumental variable is effective. If the F statistic value obtained is greater than 10, it means that the weak instrumental variable test has been passed. In models 3 and 6, the F statistical value of each variable is greater than 10, so the instrumental variables used are effective and can be used for robustness test. The results of the two-stage least squares estimation are shown in Table 4. After considering endogeneity, from the perspective of output, the coefficient of environmental regulation is positive at the 5% confidence level. In terms of cost, the coefficient of environmental regulation is positive at the 5% confidence level. The sign and significance of the core variable coefficients obtained have not changed much compared with the benchmark model, so the benchmark regression results are relatively stable. The robustness test shows that environmental regulation indeed improved the GVC position of 12 manufacturing sectors in Hebei Province.

According to the difference between capital intensity and pollution degree, manufacturing sectors are constrained by its own heterogeneity when facing environmental regulation. Therefore, this paper selects capital intensity and pollution degree as regulatory variables to further refine the impact of environmental regulation on the position of different manufacturing sectors in the GVC in Hebei Province. The econometric model is (5) and (6) as follows:(5)$${\mathrm{position}}_{\mathrm{it}}=\beta_{0}+\beta_{1}{\mathrm{regulate}}_{\mathrm{it}}+\beta_{2}{\mathrm{factor}}_{\mathrm{it}}+\beta_{3}{\;(\mathrm{regulate}}_{\mathrm{it}}{\cdot \mathrm{factor}}_{\mathrm{it}})+\beta_{4}X_{\mathrm{it}}+{\mathrm{year}}_{t}+\mu_{i}+\varepsilon_{\mathrm{it}}$$
(6)$${\mathrm{position}}_{\mathrm{it}}=\gamma_{0}+\gamma_{1}{\mathrm{regulate}}_{\mathrm{it}}+\gamma_{2}{\mathrm{percent}}_{\mathrm{it}}+\gamma_{3\;}{(\mathrm{regulate}}_{\mathrm{it}}{\cdot \mathrm{percent}}_{\mathrm{it}})+\gamma_{4}X_{\mathrm{it}}+{\mathrm{year}}_{t}+\mu_{i}+\varepsilon_{\mathrm{it}}$$

“Factor” represents the capital intensity, which is measured by the net fixed assets of the manufacturing sectors. “Percent” indicates the degree of pollution, this paper measures it by the ratio of the amount of pollution control investment in the manufacturing sector to the total amount of environmental pollution control investment in the Province; $“{\mathrm{regulate}}_{\mathrm{it}}{\cdot \mathrm{factor}}_{\mathrm{it}}”$ is the interaction between capital intensity and environmental regulation; ${”\mathrm{regulate}}_{\mathrm{it}}{\cdot \mathrm{percent}}_{\mathrm{it}}”$ is the interaction between environmental pollution and environmental regulation; $“X_{\mathrm{it}}”$ is the remaining control variable. The results are shown in Table 5 and Table 6.

As shown in Table 5, based on the perspective of output and cost, for the impact of environmental regulation on the GVC of manufacturing sectors in Hebei Province, the capital intensity is at a significant level of 5%, which means environmental regulation inhibited the improving the position of 12 manufacturing sectors in the GVC of Hebei Province. It shows that environmental regulation had different effects on GVC division position of different capital-intensive manufacturing sectors. Specifically, environmental regulation had a greater impact on the utilization of resources and a smaller impact on labor capital. Therefore, compared with labor-intensive industries, environmental regulation was more likely to affect capital-intensive industries. However, due to the characteristics of capital-intensive industries such as large scale and complex industrial structure, enterprises need to further innovate, R & D and industrial transformation and upgrading, which requires a long-time cycle. Therefore, environmental regulation was difficult to have a greater impact on GVC position of manufacturing sectors in a short period. On the contrary, the development of labor-intensive industries is relatively flexible, but their ability of anti-risk is relatively poor. Therefore, under the pressure of environmental regulation, the industry will quickly carry out internal adjustments and formulate a new f development strategy, thereby improving its GVC position.

As shown in Table 6, at a significant level of 10%, the environmental regulation promoted the position of the 12 manufacturing sectors in Hebei Province in the GVC. It shows that environmental regulation had different effects on the GVC division position of manufacturing sectors with different pollution levels, but the effect on the overall sectors is the same. To be specific, the manufacturing sectors that cause more pollution was mainly highly-polluting sectors. Therefore, under the influence of environmental regulations, in order to maintain normal operation of the industry, it should increase investment in pollution control, improve production processes, increase R & D and innovation, accelerate the reformation of industrial restructuring, and fundamentally change the characteristics of highly-polluting sectors. Therefore, environmental regulation enhanced the position of highly-polluting sectors in the GVC. On the contrary, the low-polluting industry itself was less sensitive to environmental regulations, and can comply with relevant regulations by flexibly changing strategies. Therefore, environmental regulation plays a relatively small role in improving its position in the GVC.

By distinguishing manufacturing sectors with different capital intensities and different pollution levels, Hypothesis 2 is testified in the analysis.

## 5. Conclusions and Policy Implications

### 5.1. Discussion

In this paper, the world input-output table and the domestic input-output table are nested. Although the years of the world input-output table in the world input-output database (WIOD database) are continuous, the data are only updated to 2014. Moreover, the years of the input-output table among regions in China are discontinuous. Therefore, the embedded world input-output table has the shortcomings of poor timeliness and discontinuity, resulting in fewer research years and insufficient integrity. In the future, the input-output table of each database may be consolidated to broaden the research period. At the same time, the input-output table among regions in China may have a continuous year version, which improves the timeliness and completeness of the relevant research. Hebei Province is located in the Beijing-Tianjin-Hebei region. Beijing and Tianjin have mainly developed knowledge-intensive service industries, while Hebei mainly develops manufacturing industries with low-end products. The paper didn’t test the regional heterogeneity, that is, the difference between Hebei and Beijing, and Tianjin, which have different development strategies, in terms of environmental regulation for their participation in the global value chain. It will also become one of our research directions in the future.

### 5.2. Conclusions

Based on the analyses from the perspective of output and cost, this paper con-cluded that the implementation of environmental regulation has promoted the GVC position of the 12 manufacturing sectors in Hebei Province conditional on other un-changed variables. Strict environmental regulation promoted the R & D and develop-ment capabilities of regulated enterprises. Through the reasonable transformation of technological improvements, production efficiency has been improved and the added value of products will also be increased. Thus, manufacturing sectors got more com-petitiveness in GVC, and gradually improved their GVC position. Among the control variables, human capital has a significant influence on the promotion of GVC position whilst industry scale, industry development level, and R & D ability have insignificant negative effects. The degree of openness to the international market has a significant inhibiting effect on the promotion of the GVC position in the 12 manufacturing sectors.

On the analysis of heterogeneity, manufacturing sectors were constrained by het-erogeneity when facing environmental regulation according to their own capital inten-sity and pollution level. Specifically, the capital intensity had significant negative effects on the impact of environmental regulation on the upgrading of the GVC position.

### 5.3. Policy Implications

Because Hebei Province is an indispensable component of the Beijing-Tianjin-Hebei region, and the development of the three provinces has its own priorities. Beijing and Tianjin mainly develop knowledge-intensive service industries, whilst Hebei mainly focuses on manufacturing industries with low-end products, which has led to serious environmental pollution problems. Its development is not coordinated with the resources and environment, Therefore, the impact of environmental regulation measures in Hebei Province on its participation in the global value chain should be different from that in Beijing or Tianjin. The research on environmental regulation measures in different regions is also the direction of our future research. However, the results can be used as a reference for provinces similar to Hebei that mainly develop low-end manufacturing. Therefore, based on the empirical results and analysis of this paper, the following policy implications are proposed.

First, the government should further improve the relevant policies of environmental regulation and increase the intensity of environmental regulation. Environmental regulation had a positive impact on the manufacturing industry of Hebei Province, which helped to improve its position in GVC. Specifically, long-term environmental regulation can be formulated for capital-intensive manufacturing sectors to help them adjust large-scale industrial structures. Strict environmental regulations can be formulated for manufacturing industries with high pollution to accelerate the transformation of their production process, such as the development of low emission standards for steel and related industries.

Second, enterprises should increase investment in human capital and cultivate innovative talents. Innovation ability is an important driving force for the transformation and upgrading of the manufacturing industry. Although human capital contributes to the promotion of the GVC position of the 12 manufacturing sectors in Hebei Province, its role is not significant according to the previous analysis. This shows that the talents in the manufacturing industry of Hebei Province may only account for a small part. According to the fact that the GVC position in technology and labor-intensive manufacturing sectors in Hebei Province is relatively low, the government should focus on training relevant talents in technology and labor-intensive manufacturing sectors when paying attention to the training of local innovative talents.

Third, enterprises should increase their investment in R & D to promote the upgrading of the industrial structure of the manufacturing sectors. Investment in R & D will help to increase the added value of products in the long run and help enterprises complete transformation and upgrading.

## Figures and Tables

**Figure 1 ijerph-20-02933-f001:**
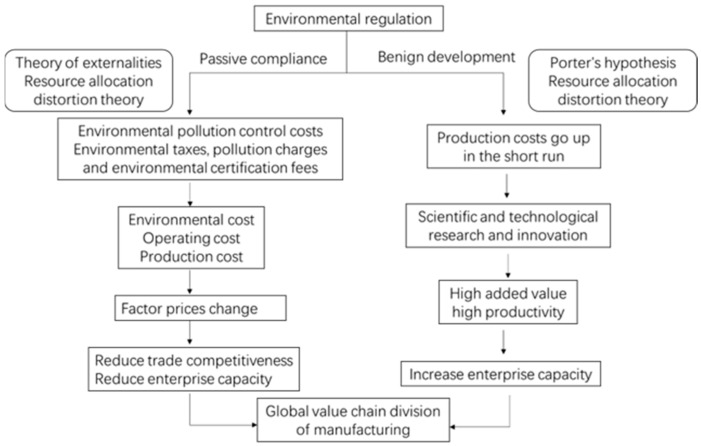
Theoretical mechanism diagram.

**Table 1 ijerph-20-02933-t001:** Description of variables and data sources.

Symbol of Variable	Name of Variable	Method of Calculation	Source of Data
*position*	Position of GVC in Hebei manufacturing sectors	GVC position index by Koopman et al., (2010)	WIOD Database, ADB database, EPS (Easy Professional Superior) Data Platform, Research Center for Virtual Economy and Data Science, Chinese Academy of Sciences, Key Laboratory of China Science Parks and Sustainable Development Analysis and Simulation, China Carbon Emission Database
*regulate*	Intensity of environmental regulation	Calculate the environmental regulatory indicators of various industries, according to the methods of Walter et al., (1973) and Dong Minjie et al., (2011) and Han Meng Meng and Yan Dongsheng et al., (2020)	China Environmental Statistical Yearbook and Hebei Economic Yearbook
*Hr*	Human capital	The ratio of the industry’s main business revenue to its average employee	China Industrial Statistics Yearbook and Hebei Economic Yearbook
*Size*	Scale of industry	The average number of employees in each industry	China Industrial Statistics Yearbook and Hebei Economic Yearbook
*Devp*	Level of industry development	The ratio of the value of sales by industry to the average number of employees	China Industrial Statistics Yearbook and Hebei Economic Yearbook
*Rd*	R & D ability	The ratio of R & D expenditure to sales value by industry	China Industrial Statistics Yearbook and Hebei Economic Yearbook
*Market*	Degree of openness to international market	The ratio of the total investment and sales value of foreign investors and Hong Kong, Macao and Taiwan in each sector	China Industrial Statistics Yearbook and Hebei Economic Yearbook

**Table 2 ijerph-20-02933-t002:** Descriptive statistical analysis of main variables.

Variable	Sample	Unit	Mean	Sd	Min	Max
*position*	60	/	0.8199	0.2632	0.3056	1.4522
*Regulate1*	60	/	0.0014	0.0011	0.0003	0.0047
*Regulate2*	60	/	0.0018	0.0015	0.0003	0.0071
*Hr*	60	one hundred million yuan/ten thousand people	81.4967	68.9287	5.6437	375.5278
*Size*	60	ten thousand people	20.9637	17.8342	1.0200	81.9400
*Devp*	60	one hundred million yuan/ten thousand people	81.4413	68.3994	8.9706	372.0703
*Rd*	60	/	0.0070	0.0049	0.0004	0.0208
*Market*	60	/	0.0324	0.0259	0.0006	0.1314

**Table 3 ijerph-20-02933-t003:** Regression results of the benchmark model.

Output Perspective				Cost Perspective			
Variables	Model 1	Model 2	Model 3	Variables	Model 4	Model 5	Model 6
*Regulate*	83.5279 *** (26.138)	66.9528 *** (34.830)	97.1734 *** (40.324)	*Regulate*	64.0239 *** (18.652)	54.0492 *** (24.4035)	69.8241 *** (27.0655)
*Hr*		0.0017 (0.009)	0.0008 (0.010)	*Hr*		0.0007 (0.009)	0.0008 (0.010)
*Size*		−0.0007 * (0.003)	−0.0006 (0.006)	*Size*		−0.0005 * (0.003)	−0.0001 (0.006)
*Devp*		−0.0012 ** (0.009)	−0.0007 ** (0.009)	*Devp*		−0.0003 ** (0.009)	−0.0008 ** (0.009)
*Rd*		−6.4633 (8.369)	−11.2832 * (10.172)	*Rd*		−6.4633 * (8.299)	−10.0907 * (9.949)
*Market*		−1.8612 * (1.551)	−2.6714 *** (1.719)	*Market*		−2.0207 * (1.539)	−2.8803 * (1.711)
Time fixed effect		Yes	Yes	Time fixed effect		Yes	Yes
Industry fixed effect			Yes	Industry fixed effect			Yes
Sample size	60	60	60	Sample size	60	60	60
R-squared	0.6123	0.6567	0.6709	R-squared	0.6201	0.6739	0.6836

Note: *, ** and *** represent the significance levels of 10%, 5% and 1%, respectively, and the numbers in the brackets are standard errors.

**Table 4 ijerph-20-02933-t004:** Estimation results of 2 SLS models.

Output Perspective		Cost Perspective	
2 SLS Result of Estimation	Model 7	2 SLS Result of Estimation	Model 8
*Regulate*	361.5992 ** (57.318)	*Regulate*	300.3727 ** (47.468)
*Hr*	0.0115 (0.009)	*Hr*	0.0131 (0.009)
*Size*	−0.0079 (0.006)	*Size*	−0.0075 (0.006)
*Devp*	−0.0115 * (0.009)	*Devp*	−0.0131 * (0.009)
*Rd*	−3.7784 * (10.0529)	*Rd*	−4.7105 * (10.096)
*Market*	−1.6629 ** (1.794)	*Market*	−1.8847 * (1.790)
Time fixed effect	Yes	Time fixed effect	Yes
Industry fixed effect	Yes	Industry fixed effect	Yes
Sample size	60	Sample size	60
R-squared	0.6822	R-squared	0.6833

Note: *, ** represent the significance levels of 10%, 5% respectively, and numbers in the brackets are standard errors.

**Table 5 ijerph-20-02933-t005:** Heterogeneity regression results from the output angle.

Variables	Model 9	Model 10	Model 11	Model 12
*Regulate*	82.8381 *** (36.952)	103.916 *** (42.381)	51.8027 *** (40.267)	60.7463 *** (45.448)
*Factor*	−0.0001 (0.001)	−0.0001 (0.000)		
*Regulate* *·factor*	−0.0229 ** (0.036)	−0.0223 ** (0.039)		
*Percent*			−22.3824 * (8.427)	−23.0730 ** (9.156)
*Regulate* *·percent*			7931.585 (3221.902)	8606.575 * (3543.651)
*Hr*	0.0022 (0.001)	0.0005 (0.0102)	0.0051 (0.009)	0.0045 (0.009)
*Size*	−0.0008 * (0.005)	−0.0014 (0.010)	−0.0044 * (0.004)	−0.0064 (0.006)
*Devp*	−0.0020 ** (0.009)	−0.0007 * (0.009)	−0.0049 ** (0.008)	−0.0045 * (0.009)
*Rd*	−6.8887 * (8.723)	−10.3268 * (10.4765)	−5.3615 ** (8.337)	−8.0685 * (1.649)
*Market*	−2.1646 (1.569)	−2.6891 ** (1.751)	−1.8613 (1.481)	−2.1646 *** (1.649)
Time fixed effect	Yes	Yes	Yes	Yes
Industry fixed effect		Yes		Yes
Sample size	60	60	60	60
R-squared	0.6729	0.6803	0.6683	0.6742

Note: *, ** and *** represent the significance levels of 10%, 5% and 1% respectively, and numbers in the brackets are standard errors.

**Table 6 ijerph-20-02933-t006:** Heterogeneity regression results from a cost perspective.

Variables	Model 13	Model 14	Model 15	Model 16
*Regulate*	61.7152 *** (25.4407)	72.5639 *** (28.240)	36.2741 *** (27.498)	38.8224 *** (30.380)
*Factor*	−0.0002 (0.000)	−0.0003 (0.000)		
*Regulate·factor*	−0.0158 ** (0.032)	−0.013 ** (0.034)		
*Percent*			−19.1115 * (7.330)	−19.7429 * (7.999)
*Regulate·percent*			5932.088 (2368.64)	6592.053 * (2628.1)
*Hr*	0.0015 (0.009)	0.0137 (0.034)	0.0041 (0.008)	0.0041 (0.009)
*Size*	−0.0012 * (0.005)	−0.0006 (0.010)	−0.0042 * (0.003)	−0.0069 (0.006)
*Devp*	−0.0013 ** (0.009)	−0.0023 * (0.009)	−0.0038 ** (0.008)	−0.0039 * (0.009)
*Rd*	−6.5837 * (8.644)	−9.2598 * (10.331)	−5.7002 ** (8.255)	−8.0365 * (9.716)
*Market*	−2.2946 (1.561)	−2.893 *** (1.747)	−1.9053 (1.476)	−2.1922 *** (1.649)
Time fixed effect	Yes	Yes	Yes	Yes
Industry fixed effect		Yes		Yes
Sample size	60	60	60	60
R-squared	0.6840	0.6897	0.6807	0.6873

Note: *, ** and *** represent the significance levels of 10%, 5% and 1% respectively, and numbers in the brackets are standard errors.

## Data Availability

Not applicable.

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
