# Peer review of "The Impact of Environmental Regulation on Hebei’s Manufacturing Industry in the Global Value Chain"

_ijerph, 2023, doi:10.3390/ijerph20042933_

Round 1

Reviewer 1 Report

Taking Hebei Province as an example, this paper discusses the impact of environmental regulation on the manufacturing industry in the global value chain. This is of great significance for China's manufacturing industry to reduce pollution and protect the environment. And to some extent, this article can also provide a reference value for China to make relevant policies to protect the environment. However, some issues with the article need to be reconsidered.

1. The research background and results of this paper are introduced in the abstract, and it is suggested that the author should add some research necessity content between the research background and results.

2. In line 48 of the introduction, the author mentions "To solve this problem, it is essential to promote the structural adjustment, Transforming and upgrading of the manufacturing industry ". This is a key point, and it is recommended that the author explain it in detail and label the source clearly.

3. This paper did not make a detailed comparison between the research results and existing literature to find the outstanding innovation points of this paper. The contribution content is very important to the results and conclusions of the empirical research.

4. It is suggested that the author add the theoretical mechanism analysis roadmap to the theoretical mechanism section, which can show the mechanism pathway more succinctly.

5. It is suggested that the author add the discussion part of the results, avoid the single description of the empirical results, and should focus on the analysis of the causes of the empirical results.

6. In terms of policy suggestions, this paper only takes the manufacturing industry of Hebei Province as the research object. Whether the empirical results have regional heterogeneity and whether the policy suggestions based on the research results are universal?

Reviewer 2 Report

Authors should improve the following items:

- Better explain the contribution of the study to the existing Literature;

- Articles should be placed in chronological order;

- Articles should be up to 5 years old by at least 50%;

- Many tables the explanation of the results should be more succinct and more objective (redo);

- Do not place two tables of results without explaining them;

- The articles used should have better Ranking (SJR: Scientific Journal Rankings; International Scientific Indexing (ISI);

- Revise the article.

Reviewer 3 Report

The research this paper constructs a fixed effect econometric model based on the panel data of the embedding level of the value chain of 12 manufacturing sectors in Hebei Province to discuss the impact of environmental regulation on the position of the value chain division. The topic is highly important, however, there are still some gaps that needs to be solved before publishing:

-In Introduction section the authors must explain better what are their contributions, both from theoretical and practical perspectives. What is new in this research? What should we do with your research?

-The Literature review needs to be updated with most recent studies like: ”On the role of institutional factors in shaping working capital management policies: Empirical evidence from European listed firms”, SG Anton, AEA Nucu, Economic Systems 46 (2), 2022.

- The selection of the variables needs to be explained. Why the authors choose these variables?

-lines “In order to solve the problem of missing variables, a two-way fixed effect (Two -way 216 FE) model incorporating time and industry is used ..” .Please mention other good papers that used the same methodology.

- Please explain better if your results are in line or not with previous findings from the literature. Also, please explain if the hypothesis are confirmed or not by the empirical results.

-Limitations of the current study and future research objectives could be mentioned at the end of the Conclusion section.

Round 2

Reviewer 2 Report

Os autores devem melhorar os seguintes itens;

- O texto da Introdução tem apenas uma aspa?!!! (você deve colocar as fontes de todas as informações);

- Os artigos devem ser colocados em ordem cronológica da Revisão da Literatura, o mesmo ocorre nesta seção demais informações sem as respectivas Fontes;

- Pouca fundamentação do estudo com base na literatura existente;

- Aprimorar o inglês técnico. 

Reviewer 3 Report

The paper shows a great improvement and, from my point of view, it can be published. 

Author Response

Thank you for reviewing our articles!